# Hydrogen Sulfide and Gut Microbiota: Their Synergistic Role in Modulating Sirtuin Activity and Potential Therapeutic Implications for Neurodegenerative Diseases

**DOI:** 10.3390/ph17111480

**Published:** 2024-11-04

**Authors:** Constantin Munteanu, Gelu Onose, Mădălina Poștaru, Marius Turnea, Mariana Rotariu, Anca Irina Galaction

**Affiliations:** 1Department of Biomedical Sciences, Faculty of Medical Bioengineering, University of Medicine and Pharmacy “Grigore T. Popa”, 700454 Iasi, Romania; madalina.postaru@umfiasi.ro (M.P.); marius.turnea@umfiasi.ro (M.T.); anca.galaction@umfiasi.ro (A.I.G.); 2Neuromuscular Rehabilitation Clinic Division, Clinical Emergency Hospital “Bagdasar-Arseni”, 041915 Bucharest, Romania; gelu.onose@umfcd.ro; 3Faculty of Medicine, University of Medicine and Pharmacy “Carol Davila”, 020022 Bucharest, Romania

**Keywords:** sirtuins, hydrogen sulfide, gut microbiota, pharmacological intervention, neurodegenerative diseases

## Abstract

The intricate relationship between hydrogen sulfide (H_2_S), gut microbiota, and sirtuins (SIRTs) can be seen as a paradigm axis in maintaining cellular homeostasis, modulating oxidative stress, and promoting mitochondrial health, which together play a pivotal role in aging and neurodegenerative diseases. H_2_S, a gasotransmitter synthesized endogenously and by specific gut microbiota, acts as a potent modulator of mitochondrial function and oxidative stress, protecting against cellular damage. Through sulfate-reducing bacteria, gut microbiota influences systemic H_2_S levels, creating a link between gut health and metabolic processes. Dysbiosis, or an imbalance in microbial populations, can alter H_2_S production, impair mitochondrial function, increase oxidative stress, and heighten inflammation, all contributing factors in neurodegenerative diseases such as Alzheimer’s and Parkinson’s. Sirtuins, particularly SIRT1 and SIRT3, are NAD^+^-dependent deacetylases that regulate mitochondrial biogenesis, antioxidant defense, and inflammation. H_2_S enhances sirtuin activity through post-translational modifications, such as sulfhydration, which activate sirtuin pathways essential for mitigating oxidative damage, reducing inflammation, and promoting cellular longevity. SIRT1, for example, deacetylates NF-κB, reducing pro-inflammatory cytokine expression, while SIRT3 modulates key mitochondrial enzymes to improve energy metabolism and detoxify reactive oxygen species (ROS). This synergy between H_2_S and sirtuins is profoundly influenced by the gut microbiota, which modulates systemic H_2_S levels and, in turn, impacts sirtuin activation. The gut microbiota–H_2_S–sirtuin axis is also essential in regulating neuroinflammation, which plays a central role in the pathogenesis of neurodegenerative diseases. Pharmacological interventions, including H_2_S donors and sirtuin-activating compounds (STACs), promise to improve these pathways synergistically, providing a novel therapeutic approach for neurodegenerative conditions. This suggests that maintaining gut microbiota diversity and promoting optimal H_2_S levels can have far-reaching effects on brain health.

## 1. Introduction

The gut microbiota, hydrogen sulfide (H_2_S), and sirtuins (SIRTs) play interconnected roles in maintaining mitochondrial health, reducing oxidative stress, and modulating inflammation, all critical for neuroprotection. Recent studies have identified a novel axis—encompassing H_2_S produced endogenously and by specific gut microbiota and modulating sirtuin activity—that may offer therapeutic potential for neurodegenerative diseases. This manuscript explores the mechanisms by which the gut microbiota influences systemic H_2_S levels and, in turn, the modulation of sirtuins, particularly SIRT1 and SIRT3, and the therapeutic implications of this axis for neuroprotection. In neurodegenerative diseases like Alzheimer’s disease (AD) and Parkinson’s disease (PD), the production of ROS surpasses the antioxidant defense, leading to cellular damage. Elevated ROS levels disrupt neuronal processes, impair protein homeostasis, and cause lipid peroxidation and DNA damage, contributing to neuron death [1].

Mitochondrial dysfunction exacerbates oxidative stress, as mitochondria are both the source and target of ROS. Impaired mitochondrial function leads to reduced ATP production, leaving neurons deprived of energy needed for signaling and synaptic function. Damaged mitochondria release pro-apoptotic factors like cytochrome c, triggering cell death, while ineffective mitophagy further aggravates mitochondrial damage, perpetuating a cycle of dysfunction and oxidative injury [2].

Inflammation also plays a critical role. Microglial activation releases pro-inflammatory cytokines in response to mitochondrial dysfunction and oxidative stress, creating a chronic neuroinflammatory state. This contributes to synaptic dysfunction, neuronal apoptosis, and blood–brain barrier (BBB) compromise, allowing peripheral immune cells to infiltrate the brain, amplifying inflammation, and promoting further neuronal damage [3]. The interplay between oxidative stress, mitochondrial dysfunction, and inflammation creates a self-perpetuating cycle that accelerates neurodegeneration. Dysfunctional mitochondria generate excessive ROS, activating microglia and astrocytes, which lead to inflammation. This chronic inflammatory state, in turn, causes further oxidative stress and mitochondrial impairment, promoting synaptic loss and the accumulation of pathological protein aggregates like amyloid-β and α-synuclein. Addressing these processes is critical to developing neuroprotective strategies [4].

Sirtuins, a family of NAD^+^-dependent deacetylases, have emerged as crucial regulators of cellular homeostasis, aging, and disease management. Their influence is also relevant in the interplay between hydrogen sulfide (H_2_S), gut microbiota, and mitochondrial health, creating a dynamic relationship with potential therapeutic implications, particularly for neurodegenerative diseases. H_2_S, produced by the host and gut microbiota, plays a critical role in activating sirtuins, such as SIRT1 and SIRT3, promoting mitochondrial biogenesis, and reducing oxidative stress, enhancing cellular longevity and resilience against dysfunction [5].

H_2_S has recently gained significant attention for its roles beyond its classical perception as a toxic gas. It is produced endogenously by enzymes such as cystathionine β-synthase (CBS), cystathionine γ-lyase (CSE), and 3-mercaptopyruvate sulfurtransferase (3-MST). It plays a neuroprotective role by mitigating oxidative damage, enhancing mitochondrial biogenesis, and modulating inflammatory responses [6]. Certain bacterial species, notably *Desulfovibrio* and *Desulfobulbus*, are critical contributors to gut-derived H_2_S through sulfate reduction pathways. These bacteria metabolize dietary sulfur compounds under anaerobic conditions, impacting systemic H_2_S levels and influencing mitochondrial function and sirtuin activation.

Moreover, emerging evidence highlights various exogenous sources of H_2_S, including balneotherapy with sulfurous waters and sapropelic mud and the dietary intake of sulfur-rich foods like garlic, onions, cruciferous vegetables, and certain supplements. H_2_S-releasing drugs, such as sodium hydrosulfide (NaHS) and GYY4137, are also being studied for their therapeutic potential. These exogenous sources contribute to the systemic pool of H_2_S, providing additional means to support cellular health and potentially offering therapeutic benefits for neurodegenerative diseases [7].

Disruptions in the gut microbiota, known as dysbiosis, can significantly impair H_2_S production, reducing sirtuin activation and compromising mitochondrial function. This dysregulation triggers oxidative stress and accelerates cellular aging, making individuals more susceptible to neurodegenerative diseases like Parkinson’s and Alzheimer’s. Restoring gut microbiota balance and, subsequently, H_2_S levels is a promising pathway to enhance sirtuin function and protect against these age-related disorders [8].

Beyond their role in mitochondrial health, sirtuins, especially SIRT1, modulate inflammation and endoplasmic reticulum (ER) stress. H_2_S plays a crucial role in boosting SIRT1 activity, particularly within the gut, where it helps maintain the intestinal barrier and reduces inflammation. This interaction suggests that modulating H_2_S production, primarily through the gut microbiota, could offer therapeutic benefits for treating inflammatory diseases such as inflammatory bowel disease (IBD) [9].

Chronic diseases like cardiovascular and neurodegenerative disorders are closely tied to mitochondrial dysfunction. SIRT1 and SIRT3 are pivotal in regulating energy metabolism and reducing oxidative damage, while gut microbiota influence H_2_S production, which supports these processes. Enhancing H_2_S signaling and restoring the gut microbial balance could thus protect against mitochondrial dysfunction in these chronic conditions [10].

In liver diseases such as non-alcoholic fatty liver disease (NAFLD), sirtuins like SIRT1 regulate lipid metabolism and inflammatory pathways, critical in preventing disease progression. Although H_2_S is not explicitly discussed in many liver disease studies, its protective roles likely overlap with sirtuin pathways. The gut microbiota, influencing liver health through the gut-liver axis, can modulate H_2_S production and sirtuin activity, offering new treatment strategies for metabolic liver disorders [11].

Sirtuins also play essential roles in inflammatory diseases. For instance, SIRT6 reduces inflammation by depleting NAD^+^ levels, a process that could be indirectly influenced by the gut microbiota. Although the connection between microbiota, H_2_S, and sirtuins is often indirect, their collective influence on inflammation and metabolism presents new avenues for therapeutic interventions, particularly in managing gut-related diseases [12].

Modulating the gut microbiota, mainly through probiotics, offers another promising approach to improving metabolic health. Probiotics can reshape microbial composition, improving glycemic control and reducing inflammation. These effects may intersect with sirtuin pathways, especially SIRT1, which regulates metabolic and inflammatory processes. By influencing H_2_S production and sirtuin activity, probiotics could provide a therapeutic strategy for managing metabolic diseases [13].

In addition to probiotics, nutraceuticals such as taurine and N-acetylcysteine have been shown to promote H_2_S synthesis, enhance sirtuin activity, and protect against oxidative stress and mitochondrial dysfunction. This approach offers insight into how gut-derived metabolites can influence sirtuin pathways, improving systemic health by reducing inflammation and oxidative stress [14].

The gut microbiota also influences epigenetic regulation, particularly in liver diseases like metabolic dysfunction-associated steatotic liver disease (MASLD). Microbiota-driven epigenetic changes, such as DNA methylation, affect genes regulated by sirtuins and H_2_S, contributing to the progression of liver diseases. Modulating the gut microbiota could enhance sirtuin activity, offering a novel therapeutic approach for managing MASLD and other liver conditions [15].

The influence of gut microbiota on sirtuins extends to neurodegenerative diseases such as Alzheimer’s, where sirtuins regulate key processes like amyloid-beta metabolism and tau deacetylation. H_2_S plays a role in reducing oxidative stress and supporting mitochondrial health, aligning with the protective effects of sirtuins in preventing neurodegeneration. Dysbiosis, or an imbalance in the gut microbial composition, can reduce sirtuin activity, potentially accelerating neurodegenerative processes [16].

Diet-induced obesity is another condition where the interplay between gut microbiota, H_2_S, and sirtuins becomes evident. H_2_S production from sulfate-reducing bacteria influences mitochondrial bioenergetics, while sirtuins like SIRT3 help protect against metabolic dysfunction by enhancing fatty acid oxidation and glucose metabolism. Dysregulation of these pathways in obesity can lead to mitochondrial dysfunction, inflammation, and metabolic disorders [17].

Disruptions in circadian rhythms can also affect the gut microbiota, H_2_S production, and sirtuin activity, impacting metabolic health. Microbial metabolites such as short-chain fatty acids (SCFAs) and H_2_S can influence peripheral clocks in organs like the liver, with sirtuins playing a role in aligning these rhythms with metabolic needs. Restoring the gut microbiota balance may thus support circadian rhythm regulation and enhance sirtuin function, improving the overall metabolic health [18].

Sirtuins also intersect with the aryl hydrocarbon receptor (AHR) pathway in aging-related tissue fibrosis. H_2_S and sirtuins regulate mitochondrial integrity and oxidative stress, with gut microbiota-derived metabolites influencing these pathways. Targeting AHR, H_2_S, and sirtuins could offer a therapeutic strategy for mitigating age-related fibrosis and organ failure [11].

## 2. Methods

This systematic review aims to evaluate the current literature on the interplay between H_2_S, gut microbiota, and sirtuins, focusing on their therapeutic potential for neurodegenerative diseases. The review follows the Preferred Reporting Items for Systematic Reviews and Meta-Analyses (PRISMA) guidelines. A comprehensive search was conducted in Google Scholar, BMJ, ClinicalKey, Nature, and SCOPUS, covering the literature from 2010 to 2024. Search terms included combinations such as “H_2_S”, “gut microbiota”, “sirtuins”, and “neurodegenerative diseases”, employing Boolean operators to refine the results. The articles were filtered based on relevance, peer-review status, and publication date. Duplicates were removed, and two reviewers independently screened the titles and abstracts for relevance. The inclusion criteria encompassed peer-reviewed experimental studies, clinical trials, reviews, meta-analyses involving animal or human subjects, and in vitro and in vivo research on H_2_S, gut microbiota, and sirtuins. Only English language studies were included. Non-peer-reviewed materials and articles unrelated to the H_2_S–gut microbiota–sirtuin axis or focused solely on other gasotransmitters were excluded (see Figure 1).

Two reviewers independently extracted data using a standardized form, collecting information on study characteristics, biological mechanisms, and therapeutic implications. Quality was assessed using the PEDro score. Discrepancies were resolved through discussion or third-party consultation.

The synthesis of the findings involved a thematic qualitative analysis, with the results organized into three domains: (1) the role of H_2_S in mitochondrial and cellular functions, (2) the regulation of H_2_S and sirtuin activity by gut microbiota, and (3) therapeutic implications for neurodegenerative diseases. Limitations of the review include potential publication bias and heterogeneity across study designs, species, and outcomes.

## 3. Results and Discussion

### 3.1. Hydrogen Sulfide: A Critical Molecule in Neuroprotection

Hydrogen sulfide (H_2_S) plays a critical role in the nervous system, contributing to neuroprotection through both endogenous and microbial production. Endogenous H_2_S is synthesized in mammalian tissues via three main enzymes: cystathionine β-synthase (CBS), cystathionine γ-lyase (CSE), and 3-mercaptopyruvate sulfurtransferase (3-MST). CBS is predominantly active in the central nervous system, while CSE is found in peripheral tissues like the cardiovascular system and liver. In mitochondria, 3-MST works alongside cysteine aminotransferase, emphasizing its role in regulating oxidative stress and energy metabolism [19].

Exogenous sources of H_2_S contribute to its physiological levels and include dietary sources, supplements, and environmental exposures. Foods rich in sulfur compounds, such as garlic, onions, and cruciferous vegetables, contain precursors that promote H_2_S production, supporting cardiovascular and neuroprotective effects. Dietary supplements like N-acetylcysteine (NAC) enhance endogenous H_2_S production by acting as cysteine precursors, while slow-releasing H_2_S donors, such as GYY4137, offer controlled delivery of H_2_S, making them viable therapeutic agents [20].

H_2_S signaling impacts vital cellular processes, including oxidative stress regulation, mitochondrial protection, and inflammatory modulation. Its antioxidant capacity is twofold: H_2_S directly scavenges reactive oxygen species (ROS) and upregulates cellular antioxidant defenses like glutathione and superoxide dismutase. This antioxidative effect is mediated by the Nrf2 pathway, enhancing cellular resilience against oxidative stress. Additionally, H_2_S inhibits NADPH oxidase activity, reducing ROS generation at its source [21].

Mitochondrial protection is another critical role of H_2_S. It enhances mitochondrial biogenesis via the activation of PGC-1α, supporting the synthesis of new mitochondria. H_2_S also optimizes mitochondrial function by sulfhydrating and protecting critical proteins in the electron transport chain (ETC), boosting ATP production efficiency. Maintaining mitochondrial membrane potential and interacting with ATP-sensitive potassium channels are other protective mechanisms exerted by H_2_S to mitigate mitochondrial dysfunction [22].

H_2_S is a potent regulator of inflammation and neuroinflammation. It produces pro-inflammatory cytokines, such as TNF-α and IL-6, and inhibits the activation of pro-inflammatory pathways like NF-κB. H_2_S maintains a balanced immune response in the brain. This anti-inflammatory activity helps protect neuronal integrity, critical in preventing neurodegeneration associated with Alzheimer’s and Parkinson’s diseases. Additionally, it inhibits the activation of inflammasomes, thereby contributing to reduced inflammation in conditions like inflammatory bowel disease (IBD) [23].

H_2_S plays a pivotal role in neuroprotection in enhancing neuronal survival and cognitive function. It protects neurons from oxidative stress by scavenging ROS and enhancing cellular antioxidant defenses. H_2_S also enhances mitochondrial health in neurons, interacting directly with the ETC to support ATP production while reducing mitochondrial depolarization. These effects collectively contribute to enhanced neuronal bioenergetics and protection from apoptosis [16].

H_2_S also supports synaptic plasticity, a fundamental feature of learning and memory, by increasing the brain-derived neurotrophic factor (BDNF) levels and modulating NMDA receptor activity. These effects enhance synaptic connectivity, improving learning and memory functions. In neurodegenerative disease models, H_2_S supplementation has been shown to reduce amyloid-beta aggregation and tau hyperphosphorylation, key markers of Alzheimer’s disease, indicating its role in maintaining cognitive function [24].

### 3.2. Gut Microbiota and Its Role in Modulating Hydrogen Sulfide Production

The gut microbiota is a diverse community of trillions of microorganisms that play a crucial role in host health, including modulating neurological function via the gut–brain axis. Comprising bacteria, viruses, fungi, and archaea, the gut microbiota influences digestion, immunity, and systemic physiology. It impacts various organs through complex interactions, including the central nervous system (CNS), forming a bidirectional communication network known as the gut–brain axis [25].

The gut microbiota’s influence on systemic health is mediated through metabolic and immune pathways. Metabolically, gut bacteria break down dietary fibers into SCFAs such as acetate, butyrate, and propionate, which regulate the gut barrier integrity, inflammation, and host energy metabolism. SCFAs also impact the CNS by crossing the BBB or stimulating the vagus nerve, thereby influencing neurochemical signaling and neuroinflammation [26].

Immune modulation by the gut microbiota is equally significant. It involves the gut-associated lymphoid tissue and interactions with immune cells to maintain systemic immune homeostasis. This immune regulation influences neuroinflammatory processes, which are critical in the context of neurodegenerative diseases like Alzheimer’s disease (AD) and Parkinson’s disease (PD) [27].

The gut–brain axis operates through neuronal, immune, and endocrine pathways, with the vagus nerve serving as the primary neural link between the gut and brain [28]. Microbial metabolites such as SCFAs can activate vagal afferents, sending direct signals to the brain that affect mood and cognition. Endocrine modulation involves neurotransmitter production by the gut bacteria, including serotonin and gamma-aminobutyric acid (GABA), which influence the hypothalamic–pituitary–adrenal (HPA) axis and regulate stress responses [29].

The gut microbiota also exerts neuroprotective effects, primarily mediated by the production of beneficial metabolites like butyrate, which reduce neuroinflammation and promote BDNF expression. These metabolites are crucial for maintaining neuronal health by supporting synaptic plasticity and cognitive function. Conversely, gut dysbiosis—an imbalance in microbial populations—can lead to increased gut permeability, systemic inflammation, and neuroinflammation, contributing to the development of neurodegenerative diseases [1].

Neuroinflammatory modulation by the gut microbiota involves the production of anti-inflammatory compounds such as SCFAs, which downregulate pro-inflammatory cytokines (e.g., TNF-α and IL-6). Beneficial bacteria, such as *Bifidobacterium* and *Lactobacillus* species, contribute to an anti-inflammatory environment by modulating immune responses and preventing excessive microglial activation in the CNS [30].

In the gut, the microbial production of H_2_S occurs through the activity of sulfate-reducing bacteria (SRB) like *Desulfovibrio* and *Desulfobulbus*. These bacteria metabolize dietary sulfur compounds such as sulfate and sulfite under anaerobic conditions. Other gut microbes, including certain *Bacteroides* and *Clostridium* species, ferment sulfur-containing amino acids like cysteine and methionine to generate H_2_S (see Figure 2). This microbial-derived H_2_S contributes to the systemic H_2_S levels, influencing mitochondrial function, redox balance, and neuronal health, forming a vital component of the H_2_S–gut microbiota–sirtuin axis. The presence of these bacteria and the subsequent production of H_2_S are significantly influenced by dietary sulfur intake, including foods like meat, eggs, and cruciferous vegetables [31].

Conversely, reduced H_2_S production due to dysbiosis can compromise mitochondrial function and increase oxidative stress, both critical in neurodegenerative processes. H_2_S is essential for maintaining redox homeostasis, supporting mitochondrial activity, and preventing neuronal apoptosis. Insufficient H_2_S limits these protective effects, increasing neuronal vulnerability [33].

The interplay between gut dysbiosis, altered H_2_S production, and reduced availability of beneficial metabolites such as SCFAs can also impair sirtuin activity, particularly SIRT1 and SIRT3, vital for mitochondrial health and energy metabolism. Dysregulated H_2_S production compromises their activation, further exacerbating mitochondrial dysfunction and promoting neurodegeneration [34].

The therapeutic potential of targeting the gut–brain axis has led to strategies involving dietary interventions, probiotics, and fecal microbiota transplantation (FMT) to restore a healthy microbiota composition. These approaches aim to enhance neuroprotection by modulating the gut microbiota, reducing neuroinflammation, and promoting neuronal survival. While H_2_S plays beneficial roles in gut health, excessive H_2_S levels can harm butyrate oxidation, compromise gut barrier integrity, and promote inflammation. Therapeutic strategies targeting the gut microbiota and modulating H_2_S production hold promise for mitigating neurodegenerative processes. Approaches like probiotics, prebiotics, and dietary interventions aim to restore the microbial balance, regulate H_2_S levels, and enhance sirtuin activation, offering a potential pathway for neuroprotection [27].

### 3.3. Sirtuins: Key Regulators of Cellular Metabolism and Neurodegeneration

In mammals, there are seven sirtuins (SIRT1–SIRT7), each with distinct cellular localizations and functions connected with diverse biological processes. Their activity is linked to the cellular energy status, as sirtuins are activated by increased NAD^+^ levels, as during caloric restriction or metabolic stress [5].

SIRT1 is the most studied member, primarily located in the nucleus. It regulates gene expression, apoptosis, and mitochondrial biogenesis by activating peroxisome proliferator-activated receptor-gamma coactivator 1-alpha (PGC-1α). SIRT1 modulates inflammation and oxidative stress by deacetylating transcription factors like FOXO3 and nuclear factor-kappa B (NF-κB), enhancing cellular stress resistance and reducing pro-inflammatory cytokine production. The neuroprotective effects of SIRT1 are well documented. In Alzheimer’s disease (AD) models, SIRT1 reduces amyloid-beta (Aβ) aggregation and tau hyperphosphorylation, key pathogenic features. By promoting synaptic plasticity and cognitive function, SIRT1 contributes to healthy neuronal aging [34].

SIRT2 is cytoplasmic, and deacetylating tubulin regulates the cell cycle, mitosis, and cytoskeletal stability. Due to its influence on the microtubule dynamics and neuronal health, it is implicated in neurodegenerative diseases like Parkinson’s disease (PD). SIRT2’s role in cellular architecture makes it a critical cell proliferation and morphology regulator [35].

SIRT3 is localized in mitochondria, where it maintains mitochondrial function by deacetylating enzymes involved in the electron transport chain (ETC), tricarboxylic acid (TCA) cycle, and fatty acid oxidation. It activates superoxide dismutase 2 (SOD2), which reduces reactive oxygen species (ROS), and plays a crucial role in maintaining redox homeostasis, preventing apoptotic cell death. SIRT3 also plays an essential role in neuroprotection, primarily through mitochondrial regulation. SIRT3 supports neuronal resilience by optimizing mitochondrial bioenergetics, particularly in conditions like PD, where mitochondrial dysfunction is prominent. SIRT1 promotes mitochondrial biogenesis, enhancing the capacity for ATP production, while SIRT3 maintains mitochondrial health and cellular energy metabolism by deacetylating vital metabolic enzymes. Together, these actions ensure mitochondrial health, critical for neuronal survival, synaptic activity, and plasticity [10].

SIRT4 also resides in mitochondria and exerts ADP-ribosyl transferase activity, regulating glutamine metabolism and influencing the cellular energy balance. Although less studied, SIRT4 has emerging roles in metabolic control, including insulin secretion and nutrient sensing, linking it to metabolic homeostasis and stress responses [5].

SIRT5 is a mitochondrial enzyme with desuccinylase, demalonylase, and deglutarylase activities. It regulates critical metabolic pathways, such as the urea cycle and antioxidant defense mechanisms. SIRT5’s regulation of ammonia detoxification and fatty acid oxidation is vital for maintaining metabolic flexibility and mitochondrial health under stress conditions [36].

SIRT6 is primarily found in the nucleus, where it regulates DNA repair, genomic stability, and inflammation. By deacetylating histones, SIRT6 maintains the chromatin structure, promotes DNA repair, and supports telomere integrity. SIRT6 modulates glucose metabolism, reducing glycolytic activity and enhancing gluconeogenesis, thereby influencing aging and cellular resilience [37].

SIRT7 is localized in the nucleolus and is involved in ribosomal biogenesis, DNA repair, and mitochondrial function. SIRT7 maintains the chromatin structure and regulates genes associated with cellular stress responses, contributing to mitochondrial homeostasis and resistance to stress [38].

Sirtuins are central to mitochondrial health, regulating both biogenesis and function. SIRT1 promotes mitochondrial biogenesis via PGC-1α, enhancing cellular energy production and resilience. SIRT3 maintains mitochondrial efficiency by optimizing ATP generation and detoxifying mitochondrial ROS, crucial for neurons that rely heavily on a mitochondrial energy supply [39].

Sirtuins’ ability to link the cellular energy status to stress responses makes them attractive targets for therapies aimed at aging and neurodegeneration. By enhancing mitochondrial biogenesis, reducing oxidative stress, and dampening inflammation, sirtuins help maintain cellular homeostasis, thereby promoting healthy aging and neuroprotection. Oxidative stress, driven by excessive ROS, contributes to neuronal damage. SIRT1 and SIRT3 mitigate oxidative damage by activating antioxidant enzymes such as SOD and catalase, reducing the ROS levels and preventing mitochondrial dysfunction [40].

The dysregulation of sirtuin activity is linked to the progression of neurodegenerative diseases. Reduced SIRT1 activity has been associated with increased Aβ deposition and tau pathology in AD, while diminished SIRT3 function contributes to impaired mitochondrial respiration and increased neuronal apoptosis. These changes exacerbate neurodegenerative processes, highlighting the importance of sirtuins in maintaining neuronal health [41].

Therapeutic approaches aimed at enhancing sirtuin activity show promise in mitigating neurodegeneration. Caloric restriction, a known activator of SIRT1 and SIRT3, has been linked to improved mitochondrial function and extended lifespan. Similarly, sirtuin-activating compounds (STACs), such as resveratrol and nicotinamide riboside (NR), have been shown to activate sirtuin pathways, reduce oxidative damage, and improve neuronal resilience [42].

### 3.4. Interplay Between Hydrogen Sulfide, Gut Microbiota, and Sirtuins

H_2_S has emerged as a significant regulator of sirtuin activity, specifically through post-translational modifications like sulfhydration. Sulfhydration involves adding a sulfur group to cysteine residues in target proteins, altering their function. This process enhances sirtuin enzymatic activities, notably those of SIRT1 and SIRT3, which play crucial roles in stress response, mitochondrial health, and inflammation regulation [43].

H_2_S-mediated sulfhydration increases SIRT1 activity, allowing it to deacetylate critical substrates involved in antioxidant defense, such as FOXO transcription factors. This upregulation enhances antioxidant enzyme expression, reducing reactive oxygen species (ROS) and protecting cells from oxidative damage. SIRT3 sulfhydration improves mitochondrial functions, deacetylating enzymes like SOD2, thereby enhancing mitochondrial resilience against stress [44].

SIRT1 plays an essential role in inflammatory regulation by deacetylating the NF-κB complex, reducing the production of inflammatory cytokines like TNF-α and IL-6. Sulfhydration by H_2_S enhances SIRT1 activity, helping mitigate chronic inflammation. This mechanism is critical for protecting neuronal cells, as neuroinflammation is a significant factor in neurodegenerative diseases such as Alzheimer’s disease (AD) and Parkinson’s disease (PD) [45].

The gut microbiota significantly influences the NAD^+^ levels, a crucial cofactor for sirtuin activity. NAD^+^ is necessary for sirtuins to function optimally, and the decline in NAD^+^ availability is linked to impaired sirtuin function and the progression of age-related diseases. The gut microbiota contributes to NAD^+^ synthesis through microbial metabolites, such as short-chain fatty acids (SCFAs), which influence NAD^+^ biosynthesis pathways. Butyrate, a SCFA produced by the gut bacteria, enhances NAD^+^ levels by modulating enzymes involved in the NAD^+^ salvage pathway. Tryptophan metabolism by gut microbes also supports NAD^+^ production through the kynurenine pathway, which is essential under inflammatory and oxidative stress conditions [46].

Certain gut bacteria produce metabolites like nicotinic acid and nicotinamide, precursors for NAD^+^ synthesis. Host cells use these metabolites to maintain NAD^+^ levels, supporting sirtuin activation. Nicotinamide riboside (NR), another NAD^+^ precursor, can also boost the NAD^+^ levels, activating sirtuin-mediated pathways that promote mitochondrial biogenesis and neuroprotection [47].

Gut dysbiosis reduces NAD^+^ availability, impairing sirtuin activity. This affects the ability of sirtuins, such as SIRT1 and SIRT3, to protect mitochondria, maintain antioxidant defenses, and inhibit inflammation. This impaired sirtuin activity contributes to mitochondrial dysfunction, increased oxidative stress, and chronic inflammation, exacerbating neurodegenerative processes [48].

The interplay between H_2_S, gut microbiota, and sirtuins forms an axis pivotal for maintaining mitochondrial health, reducing oxidative stress, and supporting neuronal survival. H_2_S promotes mitochondrial biogenesis by activating PGC-1α, essential for mitochondrial regeneration and function. It also maintains mitochondrial membrane integrity, ensuring ATP production and reducing oxidative damage [49].

Gut microbiota contributes to H_2_S production, mainly through sulfate-reducing bacteria, which supports mitochondrial function and reduces systemic oxidative stress. Moreover, SCFAs produced by gut microbiota help regulate NAD^+^ metabolism, enhancing sirtuin activation and supporting mitochondrial health and cellular resilience. Sirtuins are vital regulators of mitochondrial function. SIRT1 activates PGC-1α, promoting mitochondrial biogenesis and improving the cellular energy capacity. SIRT3 directly maintains mitochondrial integrity by regulating enzymes involved in the TCA cycle and antioxidant defenses, ensuring mitochondrial efficiency under metabolic stress [50].

H_2_S enhances SIRT1 and SIRT3 activity through sulfhydration, a modification that makes these sirtuins more effective at reducing oxidative stress and improving mitochondrial function. Enhanced sirtuin activity supports antioxidant defenses, mitochondrial energy production, and stress response pathways, all critical for preventing neuronal degeneration [51].

SIRT1’s anti-inflammatory role is bolstered by H_2_S-mediated sulfhydration, reducing NF-κB activation and pro-inflammatory cytokine production. This is essential in managing neuroinflammation, a key feature of diseases like AD and PD. SIRT6, another sirtuin, also benefits from H_2_S modification, enhancing DNA repair and maintaining genomic stability in aging neurons.

The H_2_S–gut microbiota–sirtuin axis works synergistically to promote mitochondrial biogenesis, reduce ROS, and protect neurons. H_2_S and sirtuins maintain mitochondrial integrity, vital for high-energy cells like neurons. This axis provides comprehensive neuroprotection by reducing oxidative stress, promoting mitochondrial regeneration, and inhibiting inflammation [26].

The synergistic effects of H_2_S, gut microbiota, and sirtuins have profound implications for neurodegenerative diseases (see Figure 3). Enhanced mitochondrial biogenesis and reduced oxidative stress help prevent neuronal damage and improve overall brain health, potentially slowing the progression of neurodegenerative disorders such as AD and PD [8].

Therapeutic strategies that modulate the gut microbiota to enhance H_2_S production and NAD^+^ availability, combined with sirtuin activation, offer promising avenues for treating neurodegenerative diseases. The approaches include dietary interventions, probiotics, and pharmacological agents that target these pathways to enhance mitochondrial function and neuronal resilience [52].

Dietary interventions that support SCFA production or directly increase NAD^+^ precursors can effectively modulate this axis. Additionally, pharmacological agents like H_2_S donors or sirtuin-activating compounds (e.g., resveratrol) have shown potential in activating protective mechanisms that target mitochondrial health and inflammation [53].

### 3.5. Therapeutic Implications of Targeting H_2_S, Gut Microbiota, and Sirtuins

Due to their neuroprotective properties, H_2_S donors and precursors have emerged as promising therapies for neurodegenerative diseases. H_2_S donors such as sodium hydrosulfide (NaHS) and GYY4137 release are H_2_S controlled, providing sustained protection against oxidative damage. Additionally, H_2_S donors reduce inflammation by modulating pro-inflammatory signaling pathways, contributing to neuroprotection. Precursors like L-cysteine and N-acetylcysteine (NAC) support endogenous H_2_S production, helping maintain adequate H_2_S levels, enhancing mitochondrial function, and reducing oxidative stress, thus promoting neuronal survival [54].

Modulating the gut microbiota through probiotics, prebiotics, and diet significantly enhances H_2_S production, contributing to neuroprotection. Probiotics like *Lactobacillus* and Bifidobacterium restore a healthy microbiome, indirectly promoting the activity of sulfate-reducing bacteria that produce H_2_S. Prebiotics, such as inulin, provide a nutrient source for beneficial bacteria, supporting microbial activities that enhance the H_2_S levels. Diets rich in sulfur-containing amino acids, such as garlic, onions, and cruciferous vegetables, also boost H_2_S synthesis. These strategies improve the gut microbial composition, supporting a healthy balance that influences brain health via the gut–brain axis [55].

The pharmacological activation of sirtuins, particularly SIRT1 and SIRT3, represents a promising approach for treating neurodegenerative diseases. Resveratrol, a polyphenol found in grapes and berries, enhances SIRT1 activity, promoting mitochondrial biogenesis, reducing oxidative stress, and increasing neuronal resilience. NAD^+^ precursors, such as nicotinamide riboside (NR) and nicotinamide mononucleotide (NMN), replenish the NAD^+^ levels, enhancing sirtuin activity. Increased sirtuin activity supports mitochondrial function, reduces oxidative stress, and boosts neurogenesis. Since mitochondrial dysfunction and oxidative stress are joined in neurodegenerative diseases, activating sirtuins pharmacologically can mitigate disease progression and promote neuronal survival [56].

A combined therapeutic strategy that targets H_2_S, gut microbiota, and sirtuins provides a comprehensive approach to neurodegenerative disease treatment. H_2_S donors and precursors enhance the systemic H_2_S levels, supporting cellular defenses against oxidative damage while activating sirtuins through sulfhydration. Concurrently, gut microbiota modulation through probiotics and prebiotics improves H_2_S production and maintains NAD^+^ availability, which is crucial for sirtuin activity. Sirtuin activators, like resveratrol and NAD^+^ precursors, further enhance mitochondrial health and reduce inflammation. This integrated approach targets the gut microbiota–H_2_S–sirtuin axis, creating synergistic effects that optimize redox balance, improve mitochondrial efficiency, and promote neuronal survival [57].

The interplay between gut microbiota, H_2_S, and sirtuins is central to maintaining mitochondrial health, reducing oxidative stress, and modulating inflammation. Dysregulation of this axis is implicated in neurodegenerative diseases, but interventions that enhance gut microbiota function, boost H_2_S production, and activate sirtuins can mitigate disease progression. Targeting this axis through dietary interventions, supplements, and pharmacological agents may provide comprehensive protection against neuronal degeneration and improve cognitive function [58].

H_2_S and sirtuins are both critical for mitochondrial function and anti-inflammatory responses. H_2_S supports mitochondrial health by enhancing electron transport chain efficiency and ATP production, while sirtuins like SIRT3 deacetylate key mitochondrial enzymes, optimizing energy metabolism. H_2_S also activates SIRT1 and SIRT3 via sulfhydration, enhancing their activity. These actions collectively reduce oxidative stress, improve mitochondrial resilience, and modulate inflammatory responses, providing neuroprotection against conditions such as AD, PD, and HD [59].

The gut microbiota influences the host NAD^+^ levels, which are crucial for sirtuin activation. Short-chain fatty acids (SCFAs) produced by the gut bacteria modulate NAD^+^ metabolism, supporting the activity of sirtuins like SIRT1 and SIRT3. Dysbiosis, or imbalance in the gut microbiota, can reduce the NAD^+^ levels and impair sirtuin activity, exacerbating mitochondrial dysfunction and oxidative stress in neurodegenerative diseases. NAD^+^ production can be optimized by restoring gut health, enhancing sirtuin-mediated neuroprotection [60].

Dietary interventions that promote the production of H_2_S and enhance sirtuin activity show promise in preventing neurodegeneration. Diets rich in sulfur-containing amino acids enhance H_2_S synthesis, while polyphenols like resveratrol activate SIRT1. Combined dietary strategies that boost H_2_S production and sirtuin activity can improve mitochondrial health, reduce oxidative stress, and promote neuronal survival. These approaches offer potential as non-pharmacological strategies to mitigate the effects of aging and neurodegeneration [61].

H_2_S donors and sirtuin activators work synergistically to protect against neurodegenerative diseases. H_2_S donors reduce oxidative damage, improve mitochondrial function, and activate sirtuins, while sirtuin activators enhance mitochondrial biogenesis, reduce inflammation, and support neuronal survival. Combining these therapies provides a comprehensive approach to counteracting the critical mechanisms of neurodegeneration, including oxidative stress, mitochondrial dysfunction, and chronic inflammation [62].

Recent research highlights the significant role of microbial sulfur metabolism in the human gut, which has implications for health and disease. Sulfur metabolic pathways are notably diverse, with both inorganic (e.g., sulfate) and organic sulfur sources (e.g., cysteine and taurine) contributing to H_2_S production. *Desulfovibrio* and *Bilophila* harbor genes for anaerobic sulfite reductase (Asr), which is more prevalent than dissimilatory sulfite reductase (Dsr) in the gut, suggesting an expanded capacity for H_2_S production through diverse microbial taxa. Additionally, organic sulfur metabolism, particularly from dietary sources like taurine, is widespread among gut microbes, with *Bilophila wadsworthia* and certain *Desulfovibrio* species utilizing taurine to produce H_2_S. These findings underscore the importance of organic sulfur as a substrate for gut H_2_S production and the potential influence of diet on the H_2_S levels [63].

Recent studies reveal that sulfur amino acid (SAA) restriction, primarily involving methionine and cysteine, can lead to substantial metabolic benefits, even in humans. In controlled trials, SAA restriction reduced body weight, improved fat composition, and stimulated ketone body production, showcasing a distinct metabolic profile from conventional calorie restriction. SAA restriction modifies the pathways involving oxidative stress and inflammation, indicating potential neuroprotective effects relevant to diseases like Alzheimer’s and Parkinson’s. These effects may overlap with the neuroprotective mechanisms attributed to gut microbiota-derived hydrogen sulfide (H_2_S), particularly given the interconnections between sulfur metabolism and mitochondrial health. As a part of the microbiota–sirtuin–H_2_S axis, SAA modulation might thus provide a viable pathway to leverage H_2_S’s therapeutic potential through dietary and microbiota-targeted strategies [64].

### 3.6. Discussion: Limitations, Challenges, and Future Directions

Despite the therapeutic potential of the H_2_S–gut microbiota–sirtuin axis in neurodegenerative diseases, significant gaps are in our understanding of these components and their interactions. One considerable gap lies in understanding the detailed mechanisms through which H_2_S, produced by gut microbiota, influences sirtuin activity. While sulfhydration has been shown to activate sirtuins, the exact molecular pathways remain unclear, particularly in neurodegeneration. Clarifying these pathways is essential for identifying therapeutic targets to regulate sirtuin function more effectively.

The role of specific gut bacteria in systemic H_2_S production and their influence on sirtuin activity is also underexplored (Table 1). Although a balanced microbiota is known to maintain the H_2_S levels, identifying the precise bacterial species involved and their impact on sirtuin regulation could lead to more targeted microbiome interventions. Furthermore, the influence of diet, lifestyle, and environmental factors on the H_2_S–gut microbiota–sirtuin axis is not fully understood, and the effects of long-term dietary or probiotic interventions on this axis require further study. Additionally, the roles of lesser-known sirtuin isoforms beyond SIRT1 and SIRT3 in neuroprotection remain poorly characterized. Expanding research to other sirtuins could provide a more comprehensive understanding of their neuroprotective mechanisms and offer new therapeutic opportunities.

Translating preclinical findings regarding the H_2_S–gut microbiota–sirtuin axis into effective clinical therapies faces significant challenges. The complexity and variability of the human microbiome complicate the development of standardized treatments. Factors such as age, diet, genetics, and lifestyle greatly influence microbiota composition, making it difficult to predict the outcomes of probiotic or dietary interventions across diverse populations.

Another critical challenge is optimizing the dosage and delivery methods for H_2_S donors and sirtuin activators. H_2_S has a narrow therapeutic window; it can be protective at low concentrations but harmful at higher levels. Developing delivery systems that provide a sustained, controlled release is crucial to ensure safety and efficacy. Similarly, bioavailability issues limit the effectiveness of sirtuin activators like resveratrol and NAD^+^ precursors, which require innovative formulations to enhance absorption and stability.

Long-term safety remains a concern. The consequences of the sustained modulation of the gut microbiota or prolonged sirtuin activation are still unclear. Potential adverse effects, such as gut dysbiosis or imbalances in cellular processes, must be thoroughly assessed. Additionally, neurodegenerative diseases are heterogeneous, with significant variability in pathology and progression among patients. Translating preclinical success into broad clinical efficacy is challenging, as different diseases may require tailored approaches.

Effective translation also necessitates the identification of reliable biomarkers. Biomarkers that reflect changes in H_2_S production, sirtuin activation, or microbiota composition are essential for assessing treatment efficacy in clinical trials. The absence of such biomarkers currently hinders the precise monitoring of therapeutic interventions. Future research must focus on developing reliable biomarkers to monitor H_2_S levels, sirtuin activity, and microbiota changes in real time. Identifying early indicators of neurodegenerative progression related to the H_2_S–gut microbiota–sirtuin axis could aid in early intervention and personalized treatment. Biomarkers will be instrumental in assessing the effectiveness of different therapeutic strategies and adjusting treatments accordingly.

Refining therapeutic strategies is also critical. Personalized approaches that consider individual variations in microbiota, genetics, and disease progression are needed to maximize the benefits. Innovations in drug delivery systems for H_2_S donors, sirtuin activators, and probiotics are required to overcome the current challenges. Integrating pharmacological agents with dietary interventions could also provide a holistic approach to optimizing the axis for neuroprotection.

The next step is rigorous clinical trials to establish these interventions’ efficacy, safety, and optimal doses. Randomized controlled trials (RCTs) are essential to evaluate H_2_S donors, probiotics, and sirtuin activators across different neurodegenerative diseases, considering disease-specific variations. Incorporating personalized medicine approaches by stratifying patients based on microbiome and genomic profiles will further enhance treatment efficacy. The synergistic potential of combining H_2_S donors, sirtuin activators, and microbiota-modulating interventions should be explored in clinical settings. The combined use of these agents may yield more robust neuroprotective effects than single-agent therapies, providing a comprehensive approach to targeting oxidative stress, inflammation, and mitochondrial dysfunction.

Future strategies should consider combining multiple interventions to leverage the synergistic effects of targeting the H_2_S–gut microbiota–sirtuin axis. For instance, H_2_S donors, alongside sirtuin activators and microbiota-modulating agents, could simultaneously address multiple pathogenic pathways. Such an integrated approach may be critical to achieving meaningful improvements in neuroprotection, reducing disease progression, and enhancing cognitive function. Ultimately, the goal is to establish effective, personalized therapies that harness the full potential of the H_2_S–gut microbiota–sirtuin axis. By focusing on individualized interventions, reliable biomarkers, and multidisciplinary clinical trials, the next generation of neuroprotective treatments could significantly improve the quality of life for patients with neurodegenerative diseases.

## 4. Conclusions

This review underscores the critical interplay between H_2_S, gut microbiota, and sirtuins in neuroprotection, particularly for neurodegenerative diseases such as Alzheimer’s, Parkinson’s, and Huntington’s. These elements regulate key pathological processes, including oxidative stress, mitochondrial dysfunction, and inflammation—central contributors to neurodegeneration. The synergistic roles of H_2_S, microbiota-derived metabolites, and sirtuins offer a promising avenue for mitigating the progression of these disorders.

H_2_S produced endogenously by gut microbiota also exerts significant neuroprotective effects. It acts as an antioxidant, enhances mitochondrial function, and reduces inflammation. H_2_S modulates sirtuin activity, particularly SIRT1 and SIRT3, through sulfhydration, promoting mitochondrial biogenesis and improving resilience against cellular damage.

Sirtuins, a family of NAD^+^-dependent deacetylases, play a fundamental role in maintaining mitochondrial health, reducing oxidative damage, and regulating cellular stress responses. SIRT1 and SIRT3, in particular, are critical regulators of neuronal health. Their activation by H_2_S contributes to reduced neuroinflammation and improved mitochondrial function, illustrating the interconnected nature of this neuroprotective axis.

The gut microbiota plays a pivotal role by modulating H_2_S production through sulfate-reducing bacteria and producing metabolites that influence sirtuin activity. Dysbiosis, characterized by imbalances in microbial populations, disrupts H_2_S production and diminishes sirtuin activity, thereby exacerbating oxidative stress, mitochondrial dysfunction, and inflammation—all contributors to neurodegeneration.

Targeting the H_2_S–gut microbiota–sirtuin axis represents a promising therapeutic strategy for neurodegenerative diseases. Interventions such as H_2_S donors, probiotics, sirtuin activators, and dietary modifications could help restore balance to this axis, reducing oxidative stress, improving mitochondrial health, and promoting neuronal survival. The combined use of these strategies offers a holistic approach to addressing neurodegenerative conditions’ complex, multifactorial nature.

Given the multifaceted roles of H_2_S, gut microbiota, and sirtuins, further research is essential to optimize therapeutic interventions. This includes refining delivery methods, identifying biomarkers to monitor treatment efficacy, and conducting rigorous clinical trials to validate their effectiveness. By exploring the full potential of this integrated approach, innovative therapies that slow the progression of neurodegenerative diseases and improve the quality of life for affected individuals may be possible.

Clinical applications of the H_2_S–gut microbiota–sirtuin axis could include H_2_S donors, such as sodium hydrosulfide, sirtuin-activating compounds, and probiotics, to modulate the microbiota composition. These interventions may help maintain sirtuin activity, protect against mitochondrial dysfunction, and mitigate neuroinflammation, offering a potential multi-targeted approach to treating neurodegenerative diseases. Future works should focus on optimizing these strategies and evaluating their efficacy through controlled clinical trials.

## Figures and Tables

**Figure 1 pharmaceuticals-17-01480-f001:**
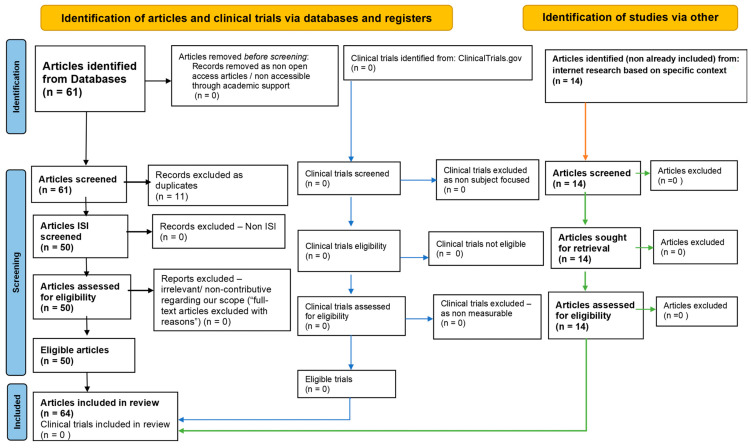
The PRISMA flow diagram used to illustrate the flow of information process.

**Figure 2 pharmaceuticals-17-01480-f002:**
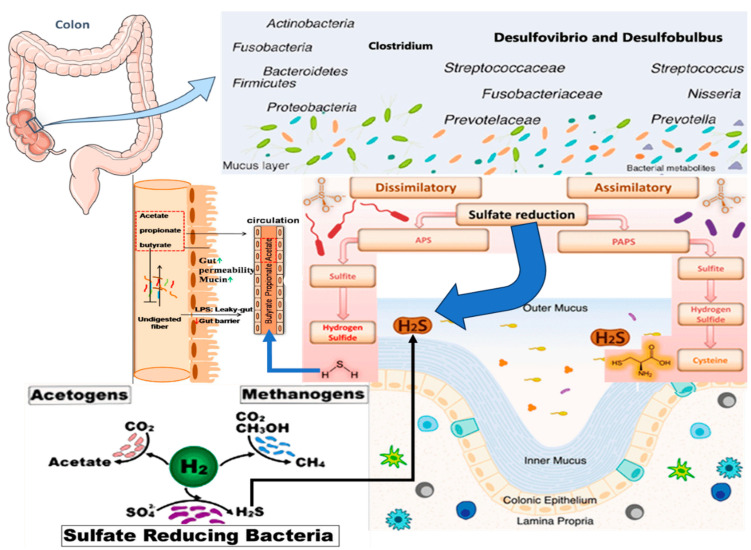
Pathways and mechanisms through which the gut microbiota, especially sulfate-reducing bacteria, produce H_2_S—steps involving sulfur compounds, such as cysteine and methionine metabolism, to clarify the microbial role in H_2_S synthesis. Gut dysbiosis can significantly alter H_2_S production, leading to pathological outcomes. An overgrowth of SRB can result in excessive H_2_S, impairing gut barrier function and contributing to systemic inflammation, which can induce neuroinflammation. The impact of dysbiosis on H_2_S production extends to its effects on the gut-brain axis, where a disrupted microbial balance influences neuroimmune and neuroinflammatory pathways. This disruption can enhance the risk of neurodegenerative diseases by increasing systemic inflammation and impairing mitochondrial function, highlighting the importance of maintaining a balanced gut microbiota for neurological health. Chronic systemic inflammation, fueled by elevated H_2_S levels, has been implicated in the pathogenesis of neurodegenerative disorders like AD and PD [32].

**Figure 3 pharmaceuticals-17-01480-f003:**
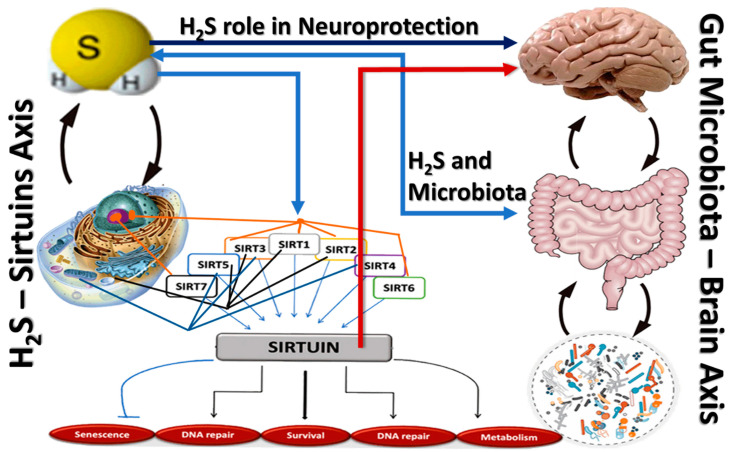
Mechanistic model of the H_2_S–gut microbiota–sirtuins axis involved in neuroprotection.

**Table 1 pharmaceuticals-17-01480-t001:** Roles of H_2_S, gut microbiota, sirtuins, and therapeutic implications in neuroprotection.

	Role of H_2_S	Role of Gut Microbiota	Role of Sirtuins	Therapeutic Implications
**Oxidative Stress**	- Acts as a scavenger of ROS, neutralizing free radicals and reducing cellular oxidative stress. - Increases the expression of antioxidant enzymes, such as glutathione peroxidase and superoxide dismutase, via activation of nuclear factor erythroid 2-related factor 2 (NRF2). - Enhances the synthesis of glutathione, a crucial cellular antioxidant, through upregulation of cysteine availability.	- Gut microbiota modulate redox balance by producing short-chain fatty acids (SCFAs) and antioxidant metabolites. - Sulfate-reducing bacteria (*Desulfovibrio*, *Desulfobacter*) help maintain physiological H_2_S levels, indirectly supporting antioxidant responses. - Dysbiosis can exacerbate oxidative stress by altering microbial metabolism and reducing beneficial ROS-scavenging compounds.	- Sirtuins, particularly SIRT1 and SIRT3, reduce oxidative damage by activating antioxidant enzymes (e.g., SOD, catalase) and deacetylating key mitochondrial proteins. - SIRT3 maintains redox homeostasis by modulating the activity of superoxide dismutase 2 (SOD2), a critical enzyme in mitochondrial ROS detoxification. - SIRT1 regulates NRF2, indirectly enhancing antioxidant response to oxidative stress.	- Use of H_2_S donors (e.g., sodium hydrosulfide) to reduce oxidative damage in neurodegenerative and cardiovascular diseases. - Sirtuin activators (e.g., resveratrol) to enhance antioxidant defenses in age-related disorders. - Probiotics targeting H_2_S-producing microbes may restore gut redox balance, potentially supporting systemic oxidative stress resilience.
**Mitochondrial Health**	- Stimulates mitochondrial biogenesis and dynamics, promoting the formation of new mitochondria to meet cellular energy demands. - Maintains electron transport chain (ETC) integrity, reducing mitochondrial ROS production and preserving cellular energy production. - Enhances ATP production efficiency by promoting mitochondrial enzyme activity, thus supporting cellular energy metabolism.	- Microbiota influence host energy metabolism and mitochondrial function by modulating the availability of H_2_S and other bioactive metabolites. - *Akkermansia muciniphila* and other beneficial microbes increase NAD^+^, an essential coenzyme for mitochondrial function and sirtuin activity. - Dysbiosis can impair mitochondrial health by altering microbial metabolites, reducing NAD^+^, and increasing ROS production.	- SIRT3 is central to mitochondrial quality control, regulating the enzymes involved in oxidative phosphorylation and fatty acid oxidation. - SIRT1 and SIRT6 promote mitochondrial biogenesis through deacetylation of peroxisome proliferator-activated receptor gamma coactivator-1 alpha (PGC-1α), a key regulator of mitochondrial production. - Sirtuins protect mitochondrial DNA (mtDNA) from damage, preserving mitochondrial integrity.	- Probiotics that increase NAD^+^ levels (e.g., *Lactobacillus plantarum*) may support mitochondrial health indirectly via sirtuin activation. - NAD^+^ precursors (e.g., nicotinamide riboside, nicotinamide mononucleotide) can activate sirtuins, promoting mitochondrial biogenesis and reducing mitochondrial dysfunction in neurodegenerative diseases. - H_2_S donors may enhance mitochondrial function, potentially treating conditions with impaired energy metabolism, like Alzheimer’s disease.
**Neuroinflammation**	- Lowers levels of pro-inflammatory cytokines (e.g., IL-1β, IL-6, TNF-α) by inhibiting nuclear factor kappa-light-chain-enhancer of activated B cells (NF-κB). - Modulates microglial activation, thus reducing neuroinflammatory responses associated with neurodegenerative diseases. - Through S-sulfhydration, H_2_S influences various signaling proteins involved in immune modulation and inflammation.	- Gut microbiota-derived H_2_S modulates immune signaling in the gut-brain axis, reducing systemic inflammation that can impact the central nervous system. - Certain gut microbes (e.g., *Bifidobacterium longum*, *Faecalibacterium prausnitzii*) produce anti-inflammatory metabolites like butyrate, contributing to an anti-inflammatory environment. - Dysbiosis may lead to chronic low-grade inflammation, negatively affecting neuroinflammatory pathways.	- SIRT1 exerts anti-inflammatory effects by deacetylating NF-κB, which reduces the transcription of pro-inflammatory cytokines. - SIRT6 modulates inflammation by controlling NF-κB signaling and inflammatory gene expression, which can ameliorate neuroinflammatory conditions. - SIRT3 contributes to an anti-inflammatory response by maintaining mitochondrial health, thus reducing mitochondrial ROS-mediated inflammatory signaling.	- Probiotics targeting neuroinflammatory pathways (e.g., *Bifidobacterium longum*) may help reduce systemic inflammation and benefit neurodegenerative disease outcomes. - H_2_S donors can regulate inflammatory signaling in the brain, providing potential therapeutic benefits in conditions like Parkinson’s disease. - Sirtuin activators (e.g., nicotinamide) are promising for managing chronic neuroinflammation in Alzheimer’s disease and other neurodegenerative conditions.

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
