# Peer review of "Hydrogen Sulfide and Gut Microbiota: Their Synergistic Role in Modulating Sirtuin Activity and Potential Therapeutic Implications for Neurodegenerative Diseases"

_pharmaceuticals, 2024, doi:10.3390/ph17111480_

Round 1

Reviewer 1 Report

Comments and Suggestions for Authors

This review explores the complex relationship between hydrogen sulfide (Hâ‚‚S), gut microbiota, and sirtuins (SIRTs), which is an interesting and relevant topic. However, the manuscript’s structure and writing need significant improvement. I recommend reconsideration for publication after revisions are made.

  1. The logic and flow of the introduction require improvement. Currently, there is little cohesion between paragraphs, making it feel disjointed and lacking a clear narrative.
  2. There are too many short paragraphs, which fragment the discussion. Consider consolidating related ideas and discussing references together to enhance readability.
  3. A table summarizing key points from the references would be beneficial, helping readers better understand the connections between Hâ‚‚S, gut microbiota, and
  4. Are there any potential clinical applications based on the Hâ‚‚S-gut microbiota-sirtuin axis? Discussing this would enhance the review’s relevance.

Comments on the Quality of English Language

Please see my last comments.

Author Response

We thank the Reviewer for the valuable feedback and recommendations to improve the clarity and flow of our manuscript, "Hydrogen Sulfide and Gut Microbiota: Their Synergistic Role in Modulating Sirtuin Activity and Potential Therapeutic Implications for Neurodegenerative Diseases." We have addressed each of the suggestions in detail, as outlined below.

1. The logic and flow of the introduction require improvement. Currently, there is little cohesion between paragraphs, making it feel disjointed and lacking a clear narrative.

Response: The introduction has been revised to establish a more cohesive narrative, guiding readers through the background and rationale for focusing on the Hâ‚‚S-gut microbiota-sirtuin axis. We have restructured paragraphs to introduce the critical roles of Hâ‚‚S, gut microbiota, and sirtuins in neuroprotection sequentially and to connect these topics more clearly to the manuscript’s overarching goals.

2. There are too many short paragraphs, which fragment the discussion. Consider consolidating related ideas and discussing references together to enhance readability.

Response: We have combined shorter paragraphs throughout the manuscript to form more comprehensive discussions. This restructuring includes consolidating related ideas and synthesizing references to provide a smoother reading experience. Each section now follows a more logical flow, reducing fragmentation and improving readability.

3. A table summarizing key points from the references would be beneficial, helping readers better understand the connections between Hâ‚‚S, gut microbiota, and sirtuins.

Response: We added a table summarizing the main findings from key references. This table outlines the roles of Hâ‚‚S, gut microbiota, and sirtuins in neuroprotection, highlighting their interactions and supporting evidence from recent studies. We believe this addition will aid readers in comprehending the complex relationships discussed in the manuscript.

4. Are there any potential clinical applications based on the Hâ‚‚S-gut microbiota-sirtuin axis? Discussing this would enhance the review’s relevance.

Response: We have expanded the discussion section to include potential clinical applications of the Hâ‚‚S-gut microbiota-sirtuin axis. Specifically, we address the therapeutic implications of Hâ‚‚S donors, sirtuin activators, and microbiota-targeting interventions as neuroprotective agents. We emphasize the clinical relevance of modulating this axis to address neurodegenerative diseases and propose future research directions to explore these therapeutic strategies further.

Reviewer 2 Report

Comments and Suggestions for Authors

The manuscript “Hydrogen Sulfide and Gut Microbiota: Their Synergistic Role in Modulating Sirtuin Activity and Potential Therapeutic Implications for Neurodegenerative Diseases” by Constantin Munteanu et al. is devoted to synergistic role sulfide and gut microbiota in modulating sirtuin activity for neurodegenerative diseases.

In my opinion, the manuscript can be accepted for publication after correcting a small number of comments.

1. The abstract is too extensive and covers a large amount of general information. The authors should focus on the key points of their work, focusing on more specific information.

2. Authors need to enter a transcript of the abbreviation "BBB" before using it.

3. The authors should make some adjustments in Figure 2. Currently, its quality is low and difficult to perceive. The hydrogen sulfide molecule is blurry; the blue, orange and yellow lines are very thick, and the yellows are very light.

4. Page 7. Before paragraph 5, there are superfluous sentences in the text - «Partea superioară a formularului» and «Partea inferioară a formularului»

Author Response

We thank the Reviewer for the positive comments and constructive feedback on our manuscript, "Hydrogen Sulfide and Gut Microbiota: Their Synergistic Role in Modulating Sirtuin Activity and Potential Therapeutic Implications for Neurodegenerative Diseases." We have carefully addressed all suggested revisions and believe these improvements have strengthened the manuscript. Please find our responses to each comment below:

  1. The abstract is too extensive and covers a large amount of general information. The authors should focus on the key points of their work, focusing on more specific information.

Response: We have revised the abstract to provide a more concise overview, emphasizing our study's key findings and specific focus areas. General information has been minimized to highlight the novel aspects of the Hâ‚‚S-gut microbiota-sirtuin axis in neurodegenerative disease therapy.

  1. Authors need to enter a transcript of the abbreviation "BBB" before using it.

Response: The full term "blood-brain barrier (BBB)" has been added before its initial use in the manuscript to improve clarity.

  1. The authors should make some adjustments in Figure 2. Currently, its quality is low and difficult to perceive. The hydrogen sulfide molecule is blurry; the blue, orange, and yellow lines are very thick, and the yellows are very light.

Response: We have modified Figure 2 to enhance its resolution and readability. The hydrogen sulfide molecule has been sharpened. 

  1. Page 7. Before paragraph 5, there are superfluous sentences in the text - "Partea superioară a formularului" and "Partea inferioară a formularului."

Response: These extraneous phrases were removed.

Reviewer 3 Report

Comments and Suggestions for Authors

The manuscript is interesting. Some points are recommended to improve originality due  to recently several documents have approached the topics included in this work (https://scholar.google.com.mx/scholar?hl=es&as_sdt=0%2C5&as_ylo=2023&as_vis=1&q=sirtuin+microbiota+neurodegeneration+h2s&btnG=).

a) Please add in introduction the relevance of some specific bacteria species for producing H2S.

b) In the section 4, I suggest for adding details about the potential production of H2S by specific bacteria to enhance the suggested action by modulation of H2S-mediated actions.

c) I suggest a additional figure showing the putative mechanisms of production of H2s by microbiota

d) In Figure 2. I suggest improve the clear relationship as in the text the neuroprotection is suggested mediated (at least in part) by sirtuin modulation. In other words, change position and arrows in the figure could improve the sense of content in the manuscript.

e) Please check the conclusions to limit to that supported by the revised information. Then, add other paragraphs suggesting implications and prospective work. Also, it is desirable the inclusion of some sentences declaring the relevance of H2S or its precursors (as well as some compounds modifying the microbiota) as potential neuroprotective agents, to improve the match of Pharmaceuticals-MDPI aims and scope (https://www.mdpi.com/journal/pharmaceuticals/about)  

Author Response

We thank the Reviewer for the additional insights and suggestions that enhanced the manuscript's relevance. The following changes have been made:

  1. a) Please add in the introduction the relevance of some specific bacterial species for producing Hâ‚‚S.

Response: We have revised the introduction to include details on specific bacteria (e.g., Desulfovibrio and Desulfobulbus species).

  1. b) In Section 4, I suggest adding details about the potential production of Hâ‚‚S by specific bacteria to enhance the suggested action by modulation of Hâ‚‚S-mediated actions.

Response: Section 4 now includes additional details on the role of sulfate-reducing bacteria and other relevant microbial species in Hâ‚‚S production. We emphasize how these bacteria's Hâ‚‚S-producing pathways interact with sirtuin-modulated processes, particularly in neuroprotective contexts.

  1. c) I suggest an additional figure showing the putative mechanisms of production of Hâ‚‚S by microbiota.

Response: A new figure has been added to illustrate the specific mechanisms through which gut microbiota, including sulfate-reducing bacteria, produce Hâ‚‚S.

  1. d) In Figure 2, I suggest improving the clear relationship as in the text the neuroprotection is suggested mediated (at least in part) by sirtuin modulation. In other words, changing position and arrows in the figure could improve the sense of content in the manuscript.

Response: Figure 2 has been revised to more clearly depict the neuroprotective role of sirtuins as mediated by Hâ‚‚S. Arrow placements and figure layout were adjusted to clarify the pathway interactions, emphasizing the centrality of sirtuin modulation in the neuroprotective effects discussed in the manuscript.

  1. e) Please check the conclusions to limit to what is supported by the revised information. Then, add other paragraphs suggesting implications and prospective work. Also, it is desirable to include some sentences declaring the relevance of Hâ‚‚S or its precursors (as well as some compounds modifying the microbiota) as potential neuroprotective agents, to improve the match with Pharmaceuticals-MDPI aims and scope.

Response: We have refined the conclusions to align with findings supported by the reviewed literature, clearly distinguishing evidence-based claims from prospective hypotheses. An additional paragraph now discusses the implications of our findings for therapeutic strategies involving Hâ‚‚S donors, sirtuin activators, and microbiota-modulating compounds as potential neuroprotective agents.